# FOIL it! Find One mismatch between Image and Language caption

## Abstract

In this paper, we aim to understand whether current language and vision (LaVi) models truly grasp the interaction between the two modalities. To this end, we propose an extension of the MS-COCO dataset, FOIL-COCO, which associates images with both correct and 'foil' captions, that is, descriptions of the image that are highly similar to the original ones, but contain one single mistake ("foil word"). We show that current LaVi models fall into the traps of this data and perform badly on three tasks: a) caption classification (correct vs. foil); b) foil word detection; c) foil word correction. Humans, in contrast, have near-perfect performance on those tasks. We demonstrate that using language cues only is not enough to deal with FOIL-COCO and that it challenges the state-of-the-art by requiring a fine-grained understanding of the relation between text and image.

## 1 Introduction

Most human language understanding is grounded in perception. There is thus growing interest in combining information from language and vision in the NLP and AI communities.

So far, the primary testbeds of Language and Vision (LaVi) models have been 'Visual Question Answering' (VQA) (e.g. Antol et al. (2015); Malinowski and Fritz (2014); Malinowski et al. (2015); Gao et al. (2015); Ren et al. (2015)) and 'Image Captioning' (IC) (e.g. Hodosh et al. (2013); Fang et al. (2015); Chen and Lawrence Zitnick (2015); Donahue et al. (2015); Karpathy and Fei-Fei (2015); Vinyals et al. (2015)). Whilst some models have seemed extremely successful

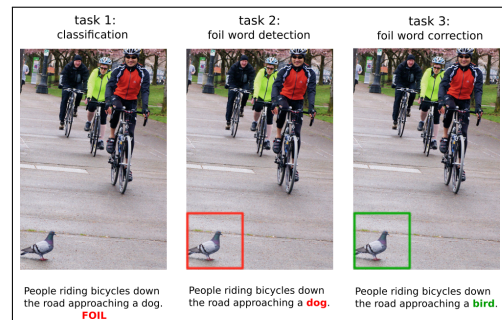

Figure 1: Is the caption correct or foil (T1)? If it is foil, where is the mistake (T2) and which is the word to correct the foil one (T3)?

on those tasks, it remains unclear how the reported results should be interpreted and what those models are actually learning. There is an emerging feeling in the LaVi community that the current VQA task should be revised, especially as it has been shown that it can be handled well by 'blind' models which use only language input or by simple concatenation of language and vision features (Agarwal et al., 2016; Jabri et al., 2016; Zhang et al., 2016; Goyal et al., 2016). In IC too, Hodosh and Hockenmaier (2016) showed that, contrarily to what prior research had suggested, the task is far from been solved, since IC models are not able to distinguish between a correct and incorrect caption.

Such results indicate that in current datasets, *language provides priors* that make it possible for a LaVi model to be successful without truly understanding and integrating language and vision. But problems do not stop at biasing. Johnson et al. (2016) highlight a second weakness of such datasets, pointing out that "they conflate multiple sources of error, making it hard to pinpoint model weaknesses". The authors thus suggest the need for a *diagnostic dataset*. Thirdly, it has been observed that existing IC *evaluation metrics* are sen-

sitive to n-gram overlap and that there is a need for measures that better substitute human judgments (Hodosh et al., 2013; Elliott and Keller, 2014; Anderson et al., 2016).

Our paper contributes to investigate these three weaknesses of current VQA and IC research by proposing an automatic method for creating a big dataset of real images with *minimal language bias* and some *diagnostic* abilities. Our dataset, called FOIL (Find One mismatch between Image and Language caption), consists of images associated with incorrect captions (Figure 1). We only introduce one error (or 'foil') per caption (i.e. the caption is 'nearly' correct). Specifically, we replace one word in the original caption with an incorrect one. This allows us to produce challenging error-detection/correction tasks while simultaneously providing ground truth (we know where the error is) that can be used to objectively measure the performance of current models. Given this data, we propose three tasks based on widely accepted evaluation measures. We evaluate a) the ability of the system to compute whether a caption is compatible (correct) or incompatible (foil) with the image (T1); b) when it is incompatible, what makes it so: the system has to highlight the mismatch in the caption (T2); c) which word should be used to correct the caption by replacing the foil word (T3). T1 is related to IC, but instead of generating a caption from scratch, we investigate the ability of the system to understand that the caption does not describe the image correctly. T2 and T3 are more related to VQA, however require a deep understanding of an image, as the aim is to spot a single mistake in an otherwise correct caption, with foils potentially close to the correct words.

The dataset presented in this paper (Section 3) is built on top of MS-COCO (Lin et al., 2014), and contains 297,268 datapoints and 97,847 images. We will refer to it as FOIL-COCO. We evaluate two state-of-the-art models: the popular VQA system proposed by Antol et al. (2015); Lu et al. (2015), and the attention-based model by Lu et al. (2016). We show that those models perform close to chance level, while humans are capable of doing the task accurately (Section 4). Section 5 provides an analysis of our results, allowing us to diagnose three failures of LaVi models. First, their coarse representations of language and visual input do not encode suitably structured information to spot mismatches between an utterance and the

corresponding scene (tested by T1). Second, their language representation is not fine-grained enough to identify the part of an utterance that causes a mismatch with the image as it is (T2). Third, their visual representation is also too poor to spot and name the visual area that corresponds to a marked mistake in the text (T3).

## 2 Related Work

Image captioning (IC) and visual question answering (VQA) tasks are the most relevant to our work. In IC (Fang et al., 2015; Chen and Lawrence Zitnick, 2015; Donahue et al., 2015; Karpathy and Fei-Fei, 2015; Vinyals et al., 2015), the goal is to generate a caption for a given image, such that it is both semantically and syntactically correct, and properly describes the content of that image. In VQA (Antol et al., 2015; Malinowski and Fritz, 2014; Malinowski et al., 2015; Gao et al., 2015; Ren et al., 2015), the system attempts to answer open-ended questions related to the content of a given image. There is a wealth of literature on both tasks, but we only discuss here the ones most related to our work and refer the reader to the recent surveys by (Bernardi et al., 2016; Wu et al., 2016).

Despite the successes of the state-of-the-art LaVi models, it is unclear whether they capture vision and language in a truly integrative fashion. We could identify three types of arguments surrounding the high performance of LaVi models:

**(i) Triviality of the LaVi tasks:** Recent work has shown that the LaVi models are mainly based on language priors (Ren et al., 2015; Agarwal et al., 2016; Kafle and Kanan, 2016) and even simple correlation and memorization (Zhou et al., 2015; Jabri et al., 2016; Hodosh and Hockenmaier, 2016) can result in a good superficial performance, without the underlying models truly understanding the visual content. Zhang et al. (2016) first unveiled that there exists a huge dataset bias in the popular VQA dataset by Antol et al. (2015): they showed that almost half of all the questions in this dataset can be answered correctly by using the question alone and ignoring the image completely. In the same vein, Zhou et al. (2015) proposed a simple baseline for the task of VQA. This baseline simply concatenates the Bag of Words (BoW) features from the question and Convolutional Neural Networks (CNN) features from the image to predict the answer. They showed that such a simple

method can achieve comparable performance to complex and deep architectures. Jabri et al. (2016) proposed a similar model for the task of multiple choice VQA, and suggested a cross-dataset generalization scheme as an evaluation criterion for VQA systems. We complement this research by introducing three new tasks with different levels of difficulty, on which LaVi models can be evaluated sequentially.

**(ii) Need for diagnostics:** To overcome the bias uncovered in previous datasets, several research groups have started proposing tasks which involve distinguishing distractors from a ground-truth caption for an image. Zhang et al. (2016) introduced a binary VQA task along with a dataset composed of sets of similar artificial images, allowing for more precise diagnostics of a system's errors. Goyal et al. (2016a) balanced the dataset of Antol et al. (2015), collecting a new set of complementary natural images which are similar to existing items in the original dataset, but result in different answers to a common question. Hodosh and Hockenmaier (2016) also proposed to evaluate a number of state-of-the-art LaVi algorithms in the presence of distractors. Their evaluation was however limited to a small dataset (namely, Flickr30K (Young et al., 2014)) and the caption generation was based on a hand-crafted scheme using only inter-dataset distractors.

Most related to our paper is the work by Ding et al. (2016).[1] Like us, they propose to extend the MS-COCO dataset by generating decoys from human-created image captions. They also suggest an evaluation apparently similar to our T1, requiring the LaVi system to detect the true target caption amongst the decoys. Our efforts, however, differ in some substantial ways. First, their technique to create incorrect captions (using BLEU to set an upper similarity threshold) is so that many of those captions will differ from the gold description in more than one respect. For instance, the caption *two elephants standing next to each other in a grass field* is associated with the decoy *a herd of giraffes standing next to each other in a dirt field* (errors: *herd*, *giraffe*, *dirt*) or with *animals are gathering next to each other in a dirt field* (error: *dirt*; infelicities: *animals* and *gathering*, which are both pragmatically odd). Clearly, the more the caption changes in the decoy, the eas-

---

[1]Please note that this paper was only published on *arxiv* a few weeks ago, at the end of December 2016. We have not yet evaluated their system against our data.

ier the task becomes. In contrast, the foil captions we propose only differ from the gold description by *one* word and are thus more challenging. Secondly, their automatic caption generation means that 'correct' descriptions can be produced, resulting in some confusion in human responses to the task. We made sure to prevent such cases, and human performance on our dataset is thus close to 100%. We note as well that our task does not require any complex instructions for the annotation, indicating that it is intuitive to human beings (see §4). Thirdly, their evaluation is a multiple-choice task, where the system has to compare all captions to understand which one is *closest* to the image. This is arguably a simpler task than the one we propose, where a caption is given and the system is asked to classify it as correct or foil: as we show in §4, detecting a *correct* caption is much easier than detecting foils. So evaluating precision on both gold and foil items is crucial.

Finally, (Johnson et al., 2016) proposed CLEVR, a dataset to aid in diagnostic evaluation of VQA systems. This dataset has been designed with the explicit goal of enabling detailed analysis of different aspects of visual reasoning, by minimizing dataset biases and providing rich ground-truth representations for both images and questions.

**(iii) Lack of objective evaluation metrics:** The evaluation of Natural Language Generation (NLG) systems is known to be a hard problem. It is further unclear whether the quality of LaVi models should be measured using the same metrics designed for language-only tasks. Elliott and Keller (2014) performed a sentence-level correlation analysis of NLG evaluation measures against expert human judgements in the context of IC. Their study revealed that most of those metrics were only weakly correlated with human judgements. In the same line of research, Anderson et al. (2016) showed that the most widely-used metrics for IC fail to capture semantic propositional content, which is an essential component of human caption evaluation. They proposed a semantic evaluation metric called SPICE, that measures how effectively image captions recover objects, attributes and the relations between them. In this paper, we tackle this problem by proposing tasks which can be evaluated based on objective metrics for classification/detection error.

## 3 Dataset

In this section, we describe how we automatically generate FOIL-COCO datapoints, viz., image, original and foil caption triples. We used the training and validation Microsoft's Common Objects in Context (MS-COCO) dataset (Lin et al., 2014) (2014 version) as our starting point. In MS-COCO, each image is described by at least five descriptions written by humans via Amazon Mechanical Turk (AMT). The images contains 91 common object categories (e.g. *dog, elephant, bird, ...* and *car, bicycle, ariplane, ...*), from 11 supercategories (*Animal*, *Vehicle*, resp.), with 82 of them having more than 5K labeled instances. In total there are 123,287 images for captions (82,783 training and 40,504 validation sets).[2]

Our data generation process consists of four main steps: 1. Generation of candidate word-pairs (target::foil) replacement lists; 2. Splitting of replacement word pairs into training and testing pairs; 3. Generation of foil captions; 4. Mining of the hardest foil caption for each image-original caption pair. The last two steps are illustrated in Figure 2.[3]

**Generation of replacement word pairs (step 1)** We want to replace one word in the original caption with a wrong word, we refer to them as *target* and *foil* word, respectively. We take target and foil words to be nouns, specifically the labels of MS-COCO categories, and we couple together words belonging to the same supercategory (e.g., bicycle::motorcycle, bicycle::car, bird::dog). In this step, we have used as our vocabulary 73 out of the 91 MS-COCO categories by leaving out those categories that were names with multi-word expressions (e.g. traffic light). We have obtained 472 target::foil word pairs.

**Splitting of replacement word pairs into training and testing (step 2)** To avoid the models learning trivial correlations due to replacement frequency, we randomly split, within each supercategory, the candidate target::foil pairs which are used to generate the captions of the training vs. test sets. We have obtained 256 pairs, built out of 72 target and 70 foil words, to generate the captions for the training set and 216 pairs, containing 73 target and 71 foil words, to generate the captions

[2] MS-COCO test set is not available for downloading.
[3] We will make our dataset available with also a validation set.

for the test set.

**Generation of foil captions (step 3)** We would like to generate foil captions by replacing only target words which refer to *visually salient objects*. To this end, given an image, we replaced only those target words that occur in more than one MS-COCO caption associated with that image. Moreover, we want to use foils which are *visually not present*, viz. that refer to visual content not present in the image. Hence, given an image, an original caption and a target word, we replace the latter only with foils that are not among the labels (objects) annotated in MS-COCO for that image. We have used the images from MS-COCO training and validation sets to generate our training and test sets, respectively. We obtained 2,229,899 and 1,097,012 captions for our training and test sets, respectively.

**Mining of the hardest foil caption for each image-original caption pair (step 4)** To eliminate possible visual-language dataset bias, for each image and original caption, out of all foil captions generated in step 3, we selected only the hardest one. For this purpose, we need to model the visual-language bias of the dataset. To this end, we use Neuraltalk[4] proposed in (Fei-Fei, 2015), one of the state-of-the-art image captioning systems, pre-trained on MS-COCO. Neuraltalk is based on an LSTM which takes as input an image and generates a sentence describing its content. We obtain a neural network $\mathcal{N}$ that implicitly represents the visual-language bias in its weights. We use $\mathcal{N}$ to approximate the conditional probability of a caption $C$ given a dataset $T$ and and an image $I$ ($P(C|I,T)$). This is obtained simply using the loss $l(C, \mathcal{N}(I))$ i.e., the error obtained by comparing the pseudo-ground truth $C$ with the sentence predicted by $\mathcal{N}$: $P(C|I,T) = 1 - l(C, \mathcal{N}(I))$. We refer to (Fei-Fei, 2015) for more details on how $l()$ is computed. Finally, $P(C|I,T)$ is used to select, for an image and an original caption, the hardest foil among all the possible foil captions, viz., the one with the highest probability according to the dataset bias learned by $\mathcal{N}$. Through this process, we obtained 197,788 and 99,480 original::foil caption pairs for the training and test sets, respectively. None of the target::foil word pairs has been filtered out by this mining process.

[4] https://github.com/karpathy/neuraltalk

| | nr. of datapoints | nr. unique images | nr. of tot. captions | nr. target::foil pairs |
|---|---|---|---|---|
| Train | 197,788 | 65,697 | 395,576 | 256 |
| Test | 99,480 | 32,150 | 198,960 | 216 |

Table 1: FOIL-COCO: a dataset consisting for each image of both a correct (original MS-COCO) caption and a wrong caption. The latter is created by replacing one of the nouns in the original caption (target word) with a foil noun.

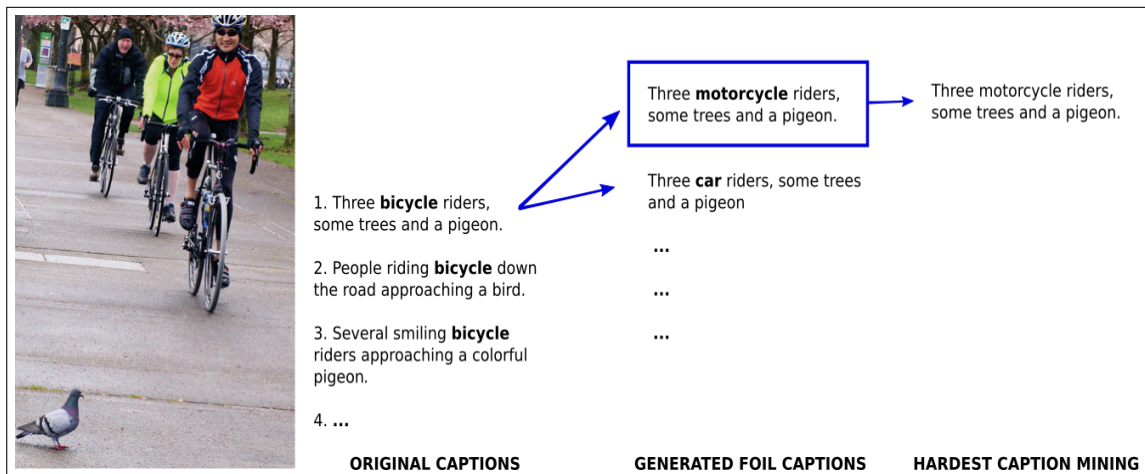

Figure 2: The main aspects of the foil caption generation process. Left column: some of the original COCO captions associated with an image. In bold we highlight one of the target words (bicycle), chosen because it is mentioned by more than one annotator. Middle column: For each original caption and each chosen target word, different foil captions are generated by replacing the target word with all possible candidate foil replacements. Right column: A single caption is selected amongst all foil candidates. We select the "hardest" caption, according to Neuraltalk model, trained using only the original captions.

The final FOIL-COCO dataset consists of 297,268 datapoints (197,788 in training and 99,480 in test set). All the 11 MS-COCO supercategories are represented in our dataset and contain 73 categories from the 91 MS-COCO ones (4.8 categories per supercategory on average.) Further details are reported in Table 1.

## 4 Experiments and Results

We have conducted three tasks aiming at understanding whether LaVi models can spot if there are mismatches between their coarse representations of language vs. visual inputs (T1); if their language representation is fine-grained enough to detect where the mismatch is (T2); if their visual representation is fine-grained enough so to be able to extract from it the information necessary to correct the foil (T3).

**Task 1 (T1): Correct vs. foil classification** Given an image and a caption, the model is asked to mark whether the caption is correct or wrong.

**Task 2 (T2): Foil word detection** Given an image and a foil caption, the model has to detect the foil word. In order to systematically check the system's performance with different prior information, for this task, we have tested two different settings (a) the foil has to be selected among only the nouns in the caption, or (b) among all the content words in it.

**Task 3 (T3): Foil word correction** Given an image, a foil caption and the foil word, the model has to detect the foil one and provide its correction. For computational reasons, we have instantiated this task by asking models to correct the foil word by selecting the correct word from the target words, instead of from the whole dataset vocabulary (viz. more that 10K words).

### 4.1 Models

There are two possible approaches to solve the proposed tasks: by using IC models and classify the top ranked captions in the generation process;

or by using VQA models. State-of-the-art IC models are trained to generate captions using Recursive Neural Network (RNN), this make them not suitable for our tasks since our foil captions contain only one wrong word. These models are instead useful for selecting over n-captions the best one (Hodosh and Hockenmaier, 2016; Ding et al., 2016). Our classification task could be seen as a multiple choice question, where there are only two choices "correct" and "foil". Therefore we have evaluated VQA models. In particular, we have used two of the three models evaluated in (Goyal et al., 2016) against a balanced VQA dataset.[5]

**LSTM + norm I:** We use the VQA best performing model in (Antol et al., 2015), namely deeper LSTM + norm I (Lu et al., 2015). This model uses a two stack Long-Short Term Memory (LSTM) to encode the questions and the last fully connected layer of VGGNet to encode the images. Both image embedding and caption embedding are projected into a 1024-dimensional feature space. Following (Goyal et al., 2016), we have normalized the image feature before projecting it into a new feature space. The combination of these two projected embeddings is performed by a point-wise multiplication. The multi-model representation so obtained is used for the classification, which is performed by multi-layer perceptron (MLP) classifier. The original VQA model is trained on 1000 most frequent answers, while in our case it is trained on only 2 possible answers i.e. "correct" or "foil" captions.

**HieCoAtt** We use the Hierarchical Co-Attention model proposed by (Lu et al., 2016) that co-attends to both the image and the question to solve the task. In particular, we evaluated the alternate version, viz. the model that sequentially alternates between generating image and question attention, and does so in a hierarchical way by starting from the word-level, then going to the phrase and then to the entire question-level. These levels are combined recursively to produce the distribution over the foil vs. correct captions.

For the *foil word detection task*, we have applied the occlusion method to the models above. Following (Goyal et al., 2016b), we systematically

occlude subsets of the language input, forward propagate the masked input through the model, and compute the change in the probability of the answer predicted with the unmasked original input. For the *error correction task*, we applied the linear regression method over all the target words and selected the target word which has the highest probability of making that wrong caption correct with respect to the given image.

**Baselines** We compare the SoA models above against the following baselines. For the classification task, we used a **Blind** model, viz. MLP+BoW (Jabri et al., 2016). This model only accepts captions as input to predict the answer. Differently from (Jabri et al., 2016) instead of concatenating BoW representations of caption and corresponding answer as input to MLP, we only input BoW representation of caption to the MLP network and use it as classifier. Apart from that, we have used the same number of parameters, viz. MLP has 2 hidden layers having 8,192 hidden units and dropout is used after the first layer. For the error detection and error classification tasks, we compare the models' results against chance.[6]

**Upper-bound** Using Crowdflower, we collected human answers from 256 subjects for 352 image and caption tuples randomly selected from the test set. We collected 1056 judgements (av. 3.9 per raters) and for each tuple, we took the answer provided by at least 2/3 annotators. Subjects were given an image and a caption and had to decide whether it was correct or wrong (T1) and if they thought it was wrong, they had to write which was the foil word (T2). Annotators are nearly perfect in classifying captions and detecting foil words. Hence, though, we have collected human answers only on a rather small subset of the test set, we believe their results are representative of how easy are the tasks for humans.

## 4.2 Results

As shown in Table 2, FOIL-COCO dataset is challenging. The 'blind', language-only model does badly on T1 with an accuracy of $43.95\%$ ($19.53\%$ on foil captions), demonstrating that language bias is minimal. The two state-of-the-art systems do significantly worse than humans on both T1 and

---

[5]We have not evaluated the Multimodal Compact Bilinear Pooling (Fukui et al., 2016) against our tasks, however in (Goyal et al., 2016) this model accuracy is higher than the HieCoAtt of 4.3% – which in the context of our results is a rather little difference.

[6]The average number of nouns per caption is 4.3 and average number of content words (viz., after removing the stop words) is 6.3. Similarly for T3, there are 72 possible target words for a given foil word.

T2. Systems show a strong bias towards correct captions and poor overall performance. They respectively only identify $34.51\%$ (LSTM +norm I) and $36.38\%$ (HieCoAtt) of the incorrect captions (T1). On the foil word detection task, they only reach $24.25\%$ and $33.69\%$ accuracy (T2). Their accuracy on foil word correction (T3) is extremely low, at $4.7\%$ and $4.21\%$ respectively. The result on T3 makes it clear that systems are unable to extract from the image representation the information needed to correct the foil: despite being told which element in the caption is wrong, they are not able to zoom into the correct part of the image to correct the foil word, or if they are, cannot name the object in that region.

## 5 Analysis

We performed a mixed-effect logistic regression analysis in order to check whether the performance of the models in T1 can be predicted by various linguistic variables (see results in Figure 3). We included: 1) semantic similarity between the original word and the foil (computed as the cosine between the two corresponding Word2Vec embeddings); 2) frequency of original word in FOIL-COCO captions; 3) frequency of the foil word in FOIL-COCO captions; 4) length of the caption (number of words). The mixed-effect model was performed to get rid of possible effects due to either object supercategory (indoor, food, vehicle, etc.) or target::foil pair (e.g., zebra::giraffe, boat::airplane, etc.). For both LSTM + norm I and HieCoAtt, Word2Vec similarity, frequency of the original word, and frequency of the foil word turned out to be highly reliable predictors of the model's response. The higher the values of these variables, the more the models tend to provide the wrong output. That is, when the foil word (e.g. *cat*) is semantically very similar to the original one (e.g. *dog*), the models tend to wrongly classify the caption as 'correct'. The same holds for frequency values. In particular, the higher the frequency of both the original word and the foil one, the more the models fail. This indicates that systems find it difficult to distinguish related concepts at the text-vision interface, and also that they may tend to be biased towards frequently occurring concepts, 'seeing them everywhere' even when they are not present in the image. Caption length turned out to be only a partially reliable predictor in the LSTM + norm I model, whereas it is a reliable predictor

```
              Estimate  Std. Error
capt_length   5.881e-03  3.135e-03  .
Word2vec     -8.924e+00  2.171e-01  ***
Target_Freq  -3.045e+01  9.713e-01  ***
Foil_Freq    -1.873e+02  1.133e+00  ***
```

Figure 3: Regression analysis: models' accuracy on Task 1. $***$ indicates significance at $p < 0.001$

in HieCoAtt. In particular, the longer the foil caption, the higher the probability that the model will wrongly label the caption as correct. Intuitively, the longer the caption, the harder for the model to spot that there is a foil word that makes the caption wrong. As revealed by the fairly high variance explained by the random effect related to target::foil pairs in the regression analysis, both models perform very well on some target::foil pairs, but fail on some others. (See Table 3 for same examples of easy/hard target::foil pairs.)

We also checked the average precision of several object detection models on the categories in our dataset. For each category (e.g. 'dog'), we calculated how likely the model was to localize that object in the images that contained it. We obtained between $25.12\%$ (Accessory) and $57.58\%$ (Animal) average precision, showing that detection on our dataset is not an easy task. However, we cannot conclude that the systems performed badly for that reason only. Indeed, there is low correlation between object detection precision and model accuracy (Pearson $-0.0363$ for the LSTM + norm I, $0.179$ for HieCoAtt on T1). While this is only an indicative result (because the models used for object detection are different from the end-to-end VQA models tested here), we can assume that other challenges lower the performance of the tested systems.

Finally, we checked whether there was any correlation between results and the position of the foil in the sentence, to ensure the models did not profit from any artefact of the data. We did not find any such correlation.

## 6 Conclusion

We have introduced FOIL-COCO, a large dataset of images associated with both correct and foil captions. The error production is automatically

| T1: Classification task | | | |
|---|---|---|---|
| | Overall | Correct | Foil |
| Blind | 43.95 | 68.36 | 19.53 |
| LSTM + norm I | 63.26 | **92.02** | 34.51 |
| HieCoAtt | 64.14 | 91.89 | **36.38** |
| Human | 91.48 | 91.67 | 91.28 |

| T2: Foil word detection task | | |
|---|---|---|
| | only nouns | all content words |
| Chance | 23.25 | 15.87 |
| LSTM + norm I | 26.32 | 24.25 |
| HieCoAtt | **38.79** | **33.69** |
| Human | | 100 |

| T3: Foil word correction task | |
|---|---|
| | all target words |
| Chance | 1.38 |
| LSTM + norm I | **4.7** |
| HieCoAtt | 4.21 |

Table 2: T1: Accuracy results on the *classification* task, relatively to all image-caption pairs (overall) and by type of caption (correct vs. foil); T2: Accuracy results on the *foil word detection* task, when the foil is known to be among the nouns only or when it is known to be among all the content words; T3: Accuracy results on the *foil word correction* task when the correct word has to be chosen among any of the target words.

| Top-5 | | Bottom-5 | | Top-5 | | Bottom-5 | |
|---|---|---|---|---|---|---|---|
| T1: LSTM + norm I | | | | T2: LSTM + norm I | | | |
| racket::glove | 100 | motorcycle::ariplane | 0 | drier::scissor | 100 | glove::skis | 0 |
| racket::kite | 97.29 | bicycle::ariplane | 0 | zebra::giraffe | 88.98 | snowboard::racket | 0 |
| couch::toilet | 97.11 | drier::scissors | 0 | boat::airplane | 87.87 | donut::apple | 0 |
| racket::skis | 95.23 | bus::ariplane | 0.35 | truck::airplane | 85.71 | glove::surfboard | 0 |
| giraffe::sheep | 95.09 | zebra:giraffe | 0.43 | train::airplane | 81.93 | spoon::bottle | 0 |
| T1: HieCoAtt | | | | T2: HieCoAtt | | | |
| tie::handbag | 100 | drier::scissor | 0 | zebra::elephant | 94.92 | direr::scissors | 0 |
| snowboard::glove | 100 | fork::glass | 0 | backpack::handbag | 94.44 | handbag::tie | 0 |
| racket::skis | 100 | handbag::tie | 0 | cow::zebra | 93.33 | broccolli:orange | 1.47 |
| racket::glove | 100 | motorcycle::airplane | 0 | bird::sheep | 93.11 | zebra::giraffe | 1.96 |
| backpack::handbag | 100 | train::airplane | 0 | orange::carrot | 92.37 | boat::airplane | 2.09 |

Table 3: Easiest and hardest target::foil pairs: T1 (caption classification) and T2 (foil word detection).

generated, but carefully thought out, making the task of spotting foils particularly challenging. By associating the dataset with a series of tasks, we allow for diagnosing various failures of current LaVi systems, from their coarse understanding of the correspondences between text and vision to their grasp of language and image structure.

Our hypothesis is that systems which, like humans, deeply integrate the language and vision modalities, should spot foil captions quite easily. The state-of-the art LaVi models we have tested fall through that test, implying that they fail to integrate the two modalities. To complete the analysis of these results, we plan to carry out a further task, namely ask the system to detect in the image the area that produces the mismatch with the foil word (the red box around the bird in Figure 1.) This extra step would allow us to fully diagnose the failure of the tested systems and confirm what is implicit in our results from task 3:

that the algorithms are unable to map particular elements of the text to their visual counterparts. We note that the addition of this extra step will move this work closer to the textual/visual explanation research (e.g., (Park et al., 2016; Selvaraju et al., 2016)). We will then have a pipeline able to not only test whether a mistake can be detected, but also whether the system can explain its decision: 'the wrong word is *dog* because the cyclists are in fact approaching a bird, there, in the image'.

LaVi models are a great success of recent research, and we are impressed by the amount of ideas, data and models produced in this stimulating and attractive area. With our work, we would like to push the community to think of ways the models can better merge language and vision rather than merely use one as a supplement to the other.

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
