# Peer review of "FOIL it! Find One mismatch between Image and Language caption"

_ACL 2017 — decision unknown_

[Official Review · Reviewer 1 · rating 4 · confidence 3]
soundness 5 · originality 5 · clarity 4 · impact 3 · substance 4 · appropriateness 5 · meaningful comparison 3 · presentation format Oral Presentation

In this work, the authors extend MS-COCO by adding an incorrect
caption to each existing caption, with only one word of difference.
The authors demonstrate that two state-of-the-art methods (one for VQA
and one for captioning) perform extremely poorly at a) determining if
a caption is fake, b) determining which word in a fake caption is
wrong, and c) selecting a replacement word for a given fake word.

This work builds upon a wealth of literature regarding the
underperformance of vision/language models relative to their apparent
capacities. I think this work makes concrete some of the big,
fundamental questions in this area: are vision/language models doing
"interesting" things, or not? The authors consider a nice mix of tasks
and models to shed light on the "broken-ness" of these settings, and
perform some insightful analyses of factors associated with model
failure (e.g., Figure 3).

My biggest concerns with the paper are similarity to Ding et al. That
being said, I do think the authors make some really good points; Ding
et al. generate similar captions, but the ones here differ by only one
word and *still* break the models -- I think that's a justifiably
fundamental difference. That observation demonstrates that Ding et
al.'s engineering is not a requirement, as this simple approach still
breaks things catastrophically.

Another concern is the use of NeuralTalk to select the "hardest"
foils.              While a clever idea, I am worried that the use of this model
creates a risk of self-reinforcement bias, i.e., NeuralTalk's biases
are now fundamentally "baked-in" to FOIL-COCO. 

I think the results section could be a bit longer, relative to the
rest of the paper (e.g. I would've liked more than one paragraph -- I
liked this part!)

Overall, I do like this paper, as it nicely builds upon some results
that highlight defficiencies in vision/language integration. In the
end, the Ding et al. similarity is not a "game-breaker," I think -- if
anything, this work shows that vision/language models are so easy to
fool, Ding et al.'s method is not even required.

Small things:

I would've liked to have seen another baseline that simply
concatenates BoW + extracted CNN features and trains a softmax
classifier over them. The "blind" model is a nice touch, but what
about a "dumb" vision+langauge baseline? I bet that would do close to
as well as the LSTM/Co-attention. That could've made the point of the
paper even stronger.

330: What is a supercategory? Is this from WordNet? Is this from COCO?
I understand the idea, but not the specifics.

397: has been -> were

494: that -> than

693: artefact -> undesirable artifacts (?)

701: I would have included a chance model in T1's table -- is 19.53%
[Line 592] a constant-prediction baseline? Is it 50% (if so, can't we
flip all of the "blind" predictions to get a better baseline?) I am
not entirely clear, and I think a "chance" line here would fix a lot
of this confusion.

719: ariplane

~~
After reading the author response...

I think this author response is spot-on. Both my concerns of NeuralTalk biases
and additional baselines were addressed, and I am confident that these can be
addressed in the final version, so I will keep my score as-is.